# Factors Associated with Vaccination Intention against the COVID-19 Pandemic: A Global Population-Based Study

**DOI:** 10.3390/vaccines10091539

**Published:** 2022-09-16

**Authors:** Junjie Huang, Sze Chai Chan, Samantha Ko, Harry H. X. Wang, Jacky Yuan, Wanghong Xu, Zhi-Jie Zheng, Hao Xue, Lin Zhang, Johnny Y. Jiang, Jason L. W. Huang, Ping Chen, Erlinda Palaganas, Pramon Viwattanakulvanid, Ratana Somrongthong, Andrés Caicedo, María de Jesús Medina-Arellano, Jill K. Murphy, Maria B. Arteaga Paredes, Mellissa Withers, Martin C. S. Wong

**Affiliations:** 1JC School of Public Health and Primary Care, Faculty of Medicine, The Chinese University of Hong Kong, Hong Kong 999077, China; 2Centre for Health Education and Health Promotion, Faculty of Medicine, The Chinese University of Hong Kong, Hong Kong 999077, China; 3School of Public Health, Sun Yat-sen University, Guangzhou 510275, China; 4The Seventh Affiliated Hospital, Sun Yat-sen University, Guangzhou 517108, China; 5Department of Epidemiology, School of Public Health, Fudan University, Shanghai 200437, China; 6Department of Global Health, School of Public Health, Peking University, Beijing 100871, China; 7Centre for Experimental Economics in Education, Shaanxi Normal University, Xi’an 710119, China; 8School of Population Medicine and Public Health, Chinese Academy of Medical Sciences & Peking Union Medical College, Beijing 100730, China; 9North Ruijin Hospital, Shanghai Jiaotong University, Shanghai 200025, China; 10Institute of Management, University of the Philippines Baguio, Baguio 2600, Philippines; 11College of Public Health Sciences, Chulalongkorn University, Bangkok 10330, Thailand; 12Escuela de Medicina, Colegio de Ciencias de la Salud COCSA, Universidad San Francisco de Quito USFQ, Quito 170901, Ecuador; 13Institute of Legal Research, National Autonomous University of Mexico (UNAM), Mexico City 04510, Mexico; 14Department of Psychiatry, Faculty of Medicine, The University of British Columbia, Vancouver, BC V6T 2A1, Canada; 15i3S-Instituto de Investigação e Inovação em Saúde, Universidade do Porto, 4200-135 Porto, Portugal; 16Department of Population and Public Health Sciences, Keck School of Medicine of USC, University of Southern California, Los Angeles, CA 90089, USA

**Keywords:** COVID-19, vaccination intention, chronic conditions, vaccine, mental health

## Abstract

Several vaccines have been developed for COVID-19 since the pandemic began. This study aimed to evaluate the factors associated with COVID-19 vaccination intention. A global survey was conducted across 26 countries from October, 2020 to December, 2021 using an online self-administered questionnaire. Demographic information, socio-economic status, and clinical information were collected. A logistic regression examined the associations between vaccine intention and factors such as perceptions and the presence of chronic physical and mental conditions. The sample included 2459 participants, with 384 participants (15.7%) expressing lower COVID-19 vaccination intent. Individuals who identified as female; belonged to an older age group; had a higher level of education; were students; had full health insurance coverage; or had a previous history of influenza vaccination were more willing to receive vaccination. Conversely, those who were working part-time, were self-employed, or were receiving social welfare were less likely to report an intention to get vaccinated. Participants with mental or physical health conditions were more unwilling to receive vaccination, especially those with sickle cell disease, cancer history within the past five years, or mental illness. Stronger vaccination intent was associated with recommendations from the government or family doctors. The presence of chronic conditions was associated with lower vaccine intention. Individuals with health conditions are especially vulnerable to health complications and may experience an increased severity of COVID-19 symptoms. Future research should evaluate the effectiveness of interventions targeting the vaccine perceptions and behaviours of at-risk groups. As such, public awareness campaigns conducted by the government and proactive endorsement from health physicians may help improve COVID-19 vaccination intention.

## 1. Introduction

The coronavirus disease 2019 (COVID-19) has had a significant global impact, causing more than 480 million confirmed cases and 6.1 million related deaths since the pandemic started in December 2019 [1]. Health services have been greatly disrupted due to a rapid increase in the burden of the disease [2]. To protect citizens from infection, governments have since implemented preventive measures to minimise social gatherings and movements [3]. However, such measures can be extremely costly from an economic and social perspective.

The health and pharmaceutical sectors developed effective vaccines to reduce incidence and disease severity, especially in vulnerable populations with chronic conditions [4]. The COVID-19 vaccines have been found to be effective in minimising the effects of COVID-19 by strengthening the immune system [5]. Despite its benefits, some individuals continue to express an unwillingness to receive the vaccine due to various reasons, such as safety and cost [6,7]. Therefore, it is imperative to examine the perceptions facilitating vaccination intention. Previous studies have been conducted that have evaluated the intention to receive the COVID-19 vaccination. However, these studies are limited to a specific country, with the findings lacking external validity and generalisability [8,9]. Moreover, the studies failed to take into consideration vulnerable subgroups within the population such as those with chronic physical or mental health conditions, who are more vulnerable to illness [8].

The current study aimed to evaluate the associations between COVID-19 vaccination intention and various factors including demographic characteristics, socio-economic background, and the presence of chronic physical and mental health conditions. Various perceptions influencing vaccination intention were also examined. The objective was to identify subgroups that may be less willing to be vaccinated.

## 2. Materials and Methods

### 2.1. Study Setting

A global survey was conducted by the Association of Pacific Rim Universities (APRU) Global Health Program to determine the factors associated with COVID-19 vaccination intention. The Association is made up of 60 leading universities of the Pacific Rim that have received worldwide recognition for their academic excellence and contributions to research. The APRU Global Health program was launched in 2007–2008 with the aim of addressing global and regional health issues through collaborative research efforts. It covers a variety of academic disciplines including non-communicable diseases, such as mental illness. Data were collected from individuals across 26 study sites, including countries and regions from (1) the Asia-Pacific region: Australia, mainland China, Hong Kong, India, Indonesia, Japan, Malaysia, New Zealand, the Philippines, Russia, South Korea, Taiwan, and Thailand; (2) Northern and Southern America: Canada, Central America, Columbia, Ecuador, Mexico, and Peru; (3) Europe: the United Kingdom, France, Germany, and Italy; and (4) the Middle East: Iraq; Saudi Arabia, and Oman.

### 2.2. Participant Recruitment

From October 2020 to December 2021, an international team consisting of more than 20 investigators distributed surveys online to the general population of each respective country or region through different channels, such as social media and website links sent through emails. Eligible study participants were invited through the investigators’ networks using snowball sampling. Individuals were eligible to participate if they were (1) aged 18 years old or above; (2) were capable of comprehending the study; and (3) provided informed consent. The data were stored securely on an online platform, with the database being password encrypted and only accessible by research personnel, to ensure the confidentiality of the collected information. The study was approved by the Survey and Behavioural Research Ethics Committee of the Chinese University of Hong Kong (SBRE-20-035), with ethical clearance obtained for all study sites.

### 2.3. Survey Instruments

An expert panel of primary care professionals, epidemiologists, and physicians was consulted for the pilot-testing and validation of the survey. The survey was available in eight languages and evaluated the association between the presence of chronic conditions and mental illnesses and perceptions related to COVID-19 vaccination intention. Participants provided information on (1) their demographic and socio-economic status: age, sex, race, years of education, type of residence, living status, work/study status, health insurance coverage, welfare benefits, and prior history of influenza vaccination; (2) the presence of chronic conditions: cardiovascular disease (examples: coronary heart disease, heart failure, cardiomyopathy), hypertension, type 2 diabetes, immunodeficiency (or taking medication, such as corticosteroid, that suppresses the immune system), chronic disease of the respiratory system (examples: asthma and chronic bronchitis), chronic liver disease, chronic kidney disease, cancer during past five years, and sickle cell disease; (3) the presence of mental illnesses: depression, mania/bipolar disorder, psychotic disorders (example: schizophrenia), anxiety disorder, post-traumatic stress disorder (PTSD), eating disorder, obsessive compulsive disorder (OCD), substance use disorder (SUD), attention-deficit hyperactivity disorder (ADHD), somatoform disorder, personality disorder, autism spectrum disorder, cognitive disorder, or dementia; and (4) their perceptions of the COVID-19 vaccine in relation to intention: country of production for the vaccine, recommendation from family doctor, recommendation from the Ministry of Health, whether the vaccine has been in use for two years or more, whether there are serious side-effects caused by the vaccine, whether the vaccine is used in other countries, the likelihood of being infected with COVID-19, the vaccine’s ease of access (availability out-of-hours or in pharmacies), if the vaccine is free of charge, whether restrictions on group gatherings or travel would be lifted if a high level of vaccination is met. The full questionnaire can be found in the Appendix A.

### 2.4. Statistical Analysis

The data were entered into the IBM Statistical Package for Social Sciences (SPSS) version 25 software to conduct statistical analyses. A descriptive analysis of study participants was performed based on their demographic details and socio-economic status. The associations between vaccination intention and the presence of chronic disease, mental illness, and perceptions related to the COVID-19 vaccine were examined. Vaccination intention was the outcome variable, whilst all other factors were considered as the explanatory variables. Potential associated factors with *p* values less than 0.20 in the bivariate analysis were included in a binary logistic regression model.

Separate binary regression models were set up to examine the factors listed above. The predictor variable, vaccination intention, was tested for association with the explanatory variables whilst controlling for potential confounding demographic and socio-economic factors. All *p* values less than 0.05 were considered statistically significant.

## 3. Results

### 3.1. Participant Characteristics

A total of 2459 responses were collected. The respondents had a mean age of 29.31, with a standard deviation of 11.07 years. Most of the respondents were female (*n* = 1526; 62.1%), were of Asian descent (*n* = 1616; 65.7%), had more than 12 years of education (*n* = 1995; 81.1%), lived in an urban area (*n* = 1559; 63.4%), lived with family (*n* = 1979; 80.5%), and worked full-time (*n* = 1076; 43.8%). More than half of the participants had full health insurance coverage (*n* = 1336; 54.3%), whilst 31% were receiving welfare benefits (*n* = 768; 31.2%). Approximately two-thirds of the respondents had a previous history of influenza vaccination. Detailed demographic and socio-economic statistics of the respondents are presented in Table 1. The number of respondents from each country, the style of the local communication, and the organisation of the national health system of the enrolled participants for major countries are listed in Appendix A.

### 3.2. Demographic and Socio-Economic Factors Associated with COVID-19 Vaccination Intention

In terms of intention, 84.4% of the respondents (*n* = 2075) were willing to receive the COVID-19 vaccine, whilst 15.7% (*n* = 384) reported no intention of getting vaccinated (Table 2). Age, as a continuous variable, was significantly associated with COVID-19 vaccination intention after adjustment with other demographic and socio-economic factors (cOR = 1.02, 95% CI: 1.01–1.04, *p* < 0.001; aOR = 1.04, 95% CI: 1.02–1.05, *p* < 0.001), indicating that the older population was more willing to receive vaccination. Participants who were female (cOR = 1.36, 95% CI: 1.09–1.70, *p* = 0.007; aOR = 1.31, 95% CI: 1.01–1.70, *p* = 0.044), had more than twelve years of education (compared to people with an education duration of nine years or less; cOR = 2.64, 95% CI: 1.69–4.12, *p* < 0.001; aOR = 2.05, 95% CI: 1.21–3.48, *p* = 0.008), lived in rural–urban fringe areas (living in urban areas as the reference group, aOR = 1.67, 95% CI: 1.03–2.71, *p* = 0.038), had full coverage health insurance (compared to people having no health insurance coverage; full coverage (public): cOR = 1.91, 95% CI: 1.43–2.56, *p* < 0.001, aOR = 1.78, 95% CI: 1.25–2.54, *p* = 0.001; full coverage (private): cOR = 2.01, 95% CI: 1.39–2.90, *p* < 0.001, aOR = 1.88, 95% CI: 1.23–2.89, *p* = 0.004), and had a previous history of influenza vaccination (cOR = 2.05, 95% CI: 1.64–2.55, *p* < 0.001; aOR = 2.26, 95% CI: 1.74–2.93, *p* < 0.001) were also found to have a higher willingness to get the COVID-19 vaccine. However, people receiving welfare benefits (cOR = 0.58, 95% CI: 0.46–0.73, *p* < 0.001; aOR = 0.55, 95% CI: 0.42–0.72, *p* < 0.001) were significantly less willing to intend to get vaccinated. Work and study status also significantly affected vaccination intention. Compared to individuals working full-time, students (aOR = 1.69, 95% CI: 1.19–2.39, *p* = 0.003) were more likely to report an intention to be vaccinated; part-time workers and self-employed individuals (cOR = 0.54, 95% CI: 0.38–0.77, *p* = 0.001; aOR = 0.61, 95% CI: 0.41–0.91, *p* = 0.015) and retired/unemployed individuals (cOR = 0.50, 95% CI: 0.33–0.76, *p* = 0.001; aOR = 0.45, 95% CI: 0.28–0.73, *p* = 0.001) were found to have a lower intention to be vaccinated. Additional analysis has been conducted to explore the difference in vaccine intention between regions; no significant difference has been found (Appendix A).

### 3.3. Associations between Physical Chronic Conditions and COVID-19 Vaccination Intention

Overall, a negative association was found between chronic physical health conditions and COVID-19 vaccination intention after adjusting for demographic and socio-economic variables (Table 3). In other words, individuals with chronic physical health conditions (cOR = 0.66, 95% CI: 0.52–0.85, *p* = 0.001; aOR = 0.68, 95% CI: 0.51–0.90, *p* = 0.008) were less willing to receive the COVID-19 vaccination. Individuals with sickle cell disease (cOR = 0.08, 95% CI: 0.05–0.15, *p* < 0.001; aOR = 0.08, 95% CI: 0.04–0.17, *p* < 0.001) were the least willing, followed by individuals with a history of cancer in the past five years (cOR = 0.11, 95% CI: 0.07–0.17, *p* < 0.001; aOR = 0.11, 95% CI: 0.06–0.19, *p* < 0.001), chronic kidney disease (cOR = 0.12, 95% CI: 0.07–0.21, *p* < 0.001; aOR = 0.17, 95% CI: 0.09–0.31, *p* < 0.001), chronic liver disease (cOR = 0.14, 95% CI: 0.08–0.24, *p* < 0.001; aOR = 0.24, 95% CI: 0.13–0.46, *p* < 0.001), immunodeficiency (cOR = 0.27, 95% CI: 0.18–0.40, *p* < 0.001; aOR = 0.40, 95% CI: 0.24–0.65, *p* < 0.001), type 2 diabetes (cOR = 0.38, 95% CI: 0.24–0.60, *p* < 0.001; aOR = 0.38, 95% CI: 0.22–0.64, *p* < 0.001), cardiovascular disease (cOR = 0.41, 95% CI: 0.24–0.69, *p* = 0.001; aOR = 0.47, 95% CI: 0.26–0.88, *p* = 0.018), chronic disease of the respiratory system (cOR = 0.45, 95% CI: 0.33–0.61, *p* < 0.001; aOR = 0.49, 95% CI: 0.34–0.71, *p* < 0.001), and hypertension (cOR = 0.55, 95% CI: 0.40–0.76, *p* < 0.001; aOR = 0.42, 95% CI: 0.28–0.63, *p* < 0.001).

### 3.4. Associations between Mental Illnesses and COVID-19 Vaccination Intention

A negative association between the presence of mental illness and vaccination intention was established (cOR = 0.38, 95% CI: 0.30–0.48, *p* < 0.001; aOR = 0.30, 95% CI: 0.22–0.41, *p* < 0.001, Table 4). Among the different mental illnesses, individuals with cognitive disorder or dementia (cOR = 0.06, 95% CI: 0.03–0.12, *p* < 0.001; aOR = 0.08, 95% CI: 0.04–0.19, *p* < 0.001) were the least likely to intend to be vaccinated, followed by individuals with personality disorders (cOR = 0.08, 95% CI: 0.05–0.14, *p* < 0.001; aOR = 0.09, 95% CI: 0.04–0.17, *p* < 0.001), psychotic disorders (cOR = 0.10, 95% CI: 0.06–0.17, *p* < 0.001; aOR = 0.09, 95% CI: 0.05–0.18, *p* < 0.001), autism spectrum disorder (cOR = 0.10, 95% CI: 0.06–0.18, *p* < 0.001; aOR = 0.10, 95% CI: 0.05–0.21, *p* < 0.001), substance abuse or addiction disorder (cOR = 0.11, 95% CI: 0.06–0.20, *p* < 0.001; aOR = 0.12, 95% CI: 0.06–0.24, *p* < 0.001), attention deficit hyperactivity disorder (cOR = 0.14, 95% CI: 0.09–0.23, *p* < 0.001; aOR = 0.15, 95% CI: 0.08–0.26, *p* < 0.001), bipolar disorder (cOR = 0.16, 95% CI: 0.10–0.25, *p* < 0.001; aOR = 0.16, 95% CI: 0.09–0.27, *p* < 0.001), obsessive-compulsive disorder (cOR = 0.16, 95% CI: 0.11–0.23, *p* < 0.001; aOR = 0.16, 95% CI: 0.09–0.26, *p* < 0.001), somatic symptom disorder (cOR = 0.23, 95% CI: 0.14–0.40, *p* < 0.001; aOR = 0.34, 95% CI: 0.18–0.63, *p* = 0.001), post-traumatic stress disorder (cOR = 0.28, 95% CI: 0.18–0.43, *p* < 0.001; aOR = 0.31, 95% CI: 0.18–0.51, *p* < 0.001), eating disorders (cOR = 0.29, 95% CI: 0.19–0.44, *p* < 0.001; aOR = 0.31, 95% CI: 0.18–0.52, *p* < 0.001), depression (cOR = 0.40, 95% CI: 0.31–0.53, *p* < 0.001; aOR = 0.34, 95% CI: 0.24–0.48, *p* < 0.001), and anxiety disorders (cOR = 0.56, 95% CI: 0.40–0.78, =<0.001; aOR = 0.50, 95% CI: 0.34–0.74, *p* < 0.001).

### 3.5. Associations between Perceptions and COVID-19 Vaccination Intention

It was found that vaccination intention was positively associated with various perceptions and characteristics of the COVID-19 vaccine (Table 5). Individuals were significantly more likely to intend to be vaccinated under the following circumstances: (1) recommendation by the Ministry of Health (aOR = 3.98, 95% CI: 3.07–5.16, *p* < 0.001); (2) recommendation from family doctor (aOR = 3.47, 95% CI: 2.67–4.50, *p* < 0.001); (3) the vaccination is free of charge (aOR = 3.04, 95% CI: 2.37–3.91, *p* < 0.001); (4) the vaccination is easy to get, i.e., available out-of-hours or in pharmacies (aOR = 2.78, 95% CI: 2.15–3.59, *p* < 0.001); (5) removal of restrictions on movement and gathering in groups if a majority of people got vaccinated (aOR = 2.32, 95% CI: 1.81–2.98, *p* < 0.001); (6) perceived risk of getting infected (aOR = 1.95, 95% CI: 1.52–2.51, *p* < 0.001); (7) the vaccination has been used in other countries (aOR = 1.79, 95% CI: 1.40–2.29, *p* < 0.001); (8) perception that the vaccination has no serious side effects (aOR = 1.44, 95% CI: 1.12–1.86, *p* = 0.004); and (9) country of production of the vaccine (aOR = 1.48, 95% CI: 1.15–1.89, *p* = 0.002).

## 4. Discussion

### 4.1. Summary of Major Findings

The aim of the study was to evaluate the association between COVID-19 vaccine intention and demographic characteristics (including socio-economic background and physical and mental health conditions) and vaccine perceptions. More than 15% of the respondents reported that they were not willing to receive the COVID-19 vaccine. The correlational analysis indicated that age, gender, years of education, study and employment status, health insurance coverage, welfare benefits, and a previous history of influenza vaccination were significantly associated with a high level of vaccine intention. Additionally, all physical and mental health conditions were associated with a low level of vaccine intention, with the following ones exhibiting significantly lower levels: (1) sickle cell disease; (2) past history of cancer; (3) cognitive disorder or dementia; and (4) personality disorders. Furthermore, respondents’ perceptions of the vaccine were associated with a high level of vaccine intention, particularly (1) recommendations from authorities and doctors; (2) the feasibility of getting the vaccine at the desired time and place; (3) whether the vaccine is free of charge.

### 4.2. Explanations and Comparisons with Previous Literature

The findings of the current study align with previous findings which demonstrated that certain demographic and socio-economic variables were predictive of vaccine intention and hesitation. Within the past literature, there has been a focus on men [10], the elderly population [11], and those with pre-existing health conditions [12], as these are considered vulnerable subgroups who might be more susceptible to contracting COVID-19. Men may suffer more dire consequences after contracting COVID-19 due to gender differences in lifestyle habits, such as higher levels of tobacco smoking and alcohol consumption [10]. Commonly reoccurring demographic and socio-economic characteristics associated with high levels of vaccine intention in our study included belonging to older age groups, higher educational attainment, being currently employed, a history of influenza vaccination, and health insurance coverage [13]. The findings are inconsistent across the literature, with some studies finding men to possess a stronger sense of vaccine intention [14], whilst others indicated that women were more likely to receive the vaccination [15]. An association between COVID-19 vaccination intention and education has previously been reported [11,16], which was further supported by this study’s finding that higher educational attainment indicated an increased likelihood of vaccine intention. This may be because individuals with a higher exposure to educational material are consequently more aware of the importance of achieving herd immunity, which encourages them to take the necessary precautions to protect themselves from COVID-19.

In our sample, the presence of chronic physical and mental health conditions was associated with vaccine hesitancy and lower levels of vaccine intention. This may be attributed to a lack of trust in the vaccine development process [17], a low perceived risk associated with the disease, as well as key concerns of vaccine safety [18]. The risk of exacerbating pre-existing health conditions because of potential side effects and unknown interactions with medication has been cited as the main reason for vaccine hesitancy [18,19]. Vulnerable populations, such as cancer patients, exhibited the same concern as the general population, with vaccine side effects being the largest predictor of vaccine hesitancy [20]. The vaccine hesitation and the lower level of vaccine intention demonstrated by this subgroup are alarming given their immunocompromised state, as the risk for COVID-19 incidence, severity, and mortality is significantly higher for those with chronic health conditions.

Wariness surrounding new vaccines is not uncommon, with individuals expressing concerns of safety or scepticism of vaccine effectiveness or the perceived risk that the disease poses. Therefore, it is imperative to bolster public confidence in vaccinations by establishing its necessity to accomplishing herd immunity to protect the wider community at large. The core perceptions that influenced higher levels of vaccine intention in this study were advice from doctors and authorities, the feasibility of receiving the vaccine at the desired location and time, and the vaccine cost. In the past literature, there have been mixed findings on an individual’s confidence in the COVID-19 vaccination in that it may vary greatly under the recommendation of authorities and healthcare professionals. Some may be more reassured given their trust in said professionals or in the healthcare system, [16,21] though others may express a strong distrust of government bodies and thus doubt the information campaigns that encourage getting vaccinated [14]. Moreover, vaccine intention is impacted by the possibility of obtaining the vaccine at one’s convenience. People living in developing countries, subgroups such as ethnic minorities or individuals with lower socio-economic backgrounds, or those living in rural areas may experience greater difficulty in accessing medical care in general and subsequently have reduced access to the vaccine. Additionally, the cost of the vaccine and whether it is costless may contribute to strengthening vaccine intention. If the vaccine is free of charge, the likelihood of improving vaccination rates increases, as individuals are encouraged to seek it out due to its availability [22]. However, in some cases, this is not a decisive factor in decision making, as individuals continue to express strong resistance and low levels of vaccine intention, despite it being readily available [23]. Previous literature has explored the difference in vaccine intention between mRNA and conventional vaccines, and it was found that the novelty of mRNA technology would reduce the acceptability of the vaccine. However, such vaccine hesitancy would be reduced with social conformity [24].

It is likely that vaccination intention was affected by the participants’ country of origin, as the style of local communication and healthcare coverage varied among countries. For instance, various fees such as laboratory testing, community isolation, and hospitalisation are covered in the Philippines and Thailand during the pandemic. In contrast, there is not a single nationwide system of health insurance in the US, which would lead to higher medical expenses for those who were not fully covered. A more detailed description of the healthcare coverage, national health systems, and styles of communication can be found in Appendix A.

### 4.3. Limitations

Various countries from different regions were involved in the survey, signifying a global collaboration from researchers in different study sites. The survey had been validated and pilot-tested by an expert panel made up of not only professionals with public health backgrounds and epidemiologists but also general practitioners. Nonetheless, there are limitations that need to be noted: (1) the response rate of the survey cannot be evaluated under the use of a consecutive sampling strategy, since the number of participants who received the survey invitation is unknown. (2) The survey responses were submitted over a span two years. The incidence and mortality rate of COVID-19 cases are time-sensitive, and it is difficult to evaluate the interaction between the perception of vaccination, various government policies, and the severity of the pandemic under different waves of COVID-19. (3) Selection bias may exist due to the use of a non-random sampling strategy, as the online nature of the survey may have made it difficult to attract older participants who are not regular internet users or cannot access the internet. Therefore, the generalisation of the study’s findings to other settings and populations in the future should be done with caution.

## 5. Conclusions

In this study, individuals with a lower intention to be vaccinated were identified with multivariable logistic regression models. The results demonstrated that people with any types of chronic physical or mental disorders were less willing to get vaccinated, especially those with sickle cell disease, cancer, and cognitive disorders. Since these groups of people often have a weaker immune system, they are more vulnerable to the serious symptoms of COVID-19. Given that participants were more willing to get vaccinated upon recommendations from the government and their family doctors, the authorities should work with the health sector to provide advice to vulnerable people with chronic conditions. Furthermore, people without health insurance coverage and people receiving welfare benefits were less willing to receive vaccination, which is possibly attributable to the cost of the vaccination or the difficulty of making time for the vaccination after long working hours. It is recommended that the vaccination is provided free of charge and that vaccination centres are set up in more convenient locations with later service hours to address these barriers. Since some participants were concerned about the safety of the vaccine in terms of the country of production, serious side effects, and the use of the vaccination in other countries, it is imperative that governments provide ample and factual information on the vaccination to the general public for reference. Further research exploring incentives to motivate people to get vaccinated could be beneficial to ensure that vulnerable groups are protected.

## Figures and Tables

**Table 1 vaccines-10-01539-t001:** Socio-demographic characteristics of the participants.

Variables	All Participants (*n* = 2459)	Participants Who Are Willing to Receive COVID-19 Vaccination (*n* = 2075)	Participants Who Are Unwilling to Receive COVID-19 Vaccination (*n* = 384)	*p*
**Age years** (mean ± sd)	29.31 ± 11.07	29.72 ± 11.41	27.11 ± 8.69	<0.001
**Sex *n* (%)**				
Male	886	725 (81.8%)	161 (18.2%)	0.007
Female	1526	1312 (86.0%)	214 (14.0%)	
**Race**				
Asian	1616	1357 (84.0%)	259 (16.0%)	0.001
White	252	196 (77.8%)	56 (22.2%)	
Black	167	142 (85.0%)	25 (15.0%)	
American Indian or Alaska Native	85	74 (87.1%)	11 (12.9%)	
Others	325	294 (90.5%)	31 (9.5%)	
**Years of education**				
0–9 years	100	70 (70.0%)	30 (30.0%)	<0.001
10–12 years	353	280 (79.3%)	73 (20.7%)	
>12 years	1995	1716 (86.0%)	279 (14.0%)	
**Residence**				
Urban area	1559	1313 (84.2%)	246 (15.8%)	0.122
Rural area	618	511 (82.7%)	107 (17.3%)	
Rural–urban fringe	226	200 (88.5%)	26 (11.5%)	
**Living status**				
Live alone	261	218 (83.5%)	43 (16.5%)	0.471
Live with family	1979	1681 (84.9%)	298 (15.1%)	
Live with other people	193	158 (81.9%)	35 (18.1%)	
**Work/study status**				
Full-time employed	1076	925 (86.0%)	151 (14.0%)	<0.001
Part-time/self employed	238	183 (76.9%)	55 (23.1%)	
Students	984	845 (85.9%)	139 (14.1%)	
Others	151	114 (75.5%)	37 (24.5%)	
**Health insurance coverage**				
None	518	411 (79.3%)	107 (20.7%)	<0.001
Partial coverage	536	423 (78.9%)	113 (21.1%)	
Full coverage (public)	918	808 (88.0%)	110 (12.0%)	
Full coverage (private)	418	370 (88.5%)	48 (11.5%)	
**Welfare benefits**				
Yes	768	608 (79.2%)	160 (20.8%)	<0.001
No	1681	1458 (86.7%)	223 (13.3%)	
**Previous history of influenza vaccination**				
Yes	1594	1400 (87.8%)	194 (12.2%)	<0.001
No	855	666 (77.9%)	189 (22.1%)	

**Table 2 vaccines-10-01539-t002:** Socio-demographic factors associated with COVID-19 vaccination intention.

	Univariable Analysis cOR (95% CI)	*p*	Multivariable Analysis aOR (95% CI)	*p*
**Age years**	1.02 (1.01–1.04)	<0.001	1.04 (1.02–1.05)	<0.001
**Sex**				
Male (ref)	1 (ref)		1 (ref)	
Female	1.36 (1.09–1.70)	0.007	1.31 (1.01–1.70)	0.044
**Race**				
Asian (ref)	1 (ref)	0.001	1 (ref)	0.228
White	0.67 (0.48–0.92)	0.015	0.99 (0.66–1.49)	0.960
Black	1.08 (0.69–1.69)	0.722	1.17 (0.69–1.97)	0.557
American Indian or Alaska Native	1.28 (0.67–2.45)	0.449	1.78 (0.85–3.76)	0.129
Others	1.81 (1.22–2.68)	0.003	1.54 (0.97–2.44)	0.067
**Years of education**				
0–9 years (ref)	1 (ref)	<0.001	1 (ref)	0.024
10–12 years	1.64 (1.00–2.71)	0.051	1.72 (0.97–3.05)	0.066
>12 years	2.64 (1.69–4.12)	<0.001	2.05 (1.21–3.48)	0.008
**Residence**				
Urban area (ref)	1 (ref)	0.125	1 (ref)	0.082
Rural area	0.89 (0.70–1.15)	0.381	1.20 (0.89–1.62)	0.221
Rural–urban fringe	1.44 (0.94–2.22)	0.096	1.67 (1.03–2.71)	0.038
**Living status**				
Live alone (ref)	1 (ref)	0.472	1 (ref)	0.540
Live with family	1.11 (0.78–1.58)	0.549	1.05 (0.71–1.55)	0.821
Live with other people	0.89 (0.54–1.45)	0.643	0.82 (0.47–1.43)	0.484
**Work/study status**				
Full-time (ref)	1 (ref)	<0.001	1 (ref)	<0.001
Part-time/self employed	0.54 (0.38–0.77)	0.001	0.61 (0.41–0.91)	0.015
Students	0.99 (0.77–1.27)	0.952	1.69 (1.19–2.39)	0.003
Others	0.50 (0.33–0.76)	0.001	0.45 (0.28–0.73)	0.001
**Health insurance coverage**				
None (ref)	1 (ref)	<0.001	1 (ref)	<0.001
Partial coverage	0.97 (0.72–1.31)	0.865	0.98 (0.69–1.38)	0.887
Full coverage (public)	1.91 (1.43–2.56)	<0.001	1.78 (1.25–2.54)	0.001
Full coverage (private)	2.01 (1.39–2.90)	<0.001	1.88 (1.23–2.89)	0.004
**Welfare benefits**				
Yes	0.58 (0.46–0.73)	<0.001	0.55 (0.42–0.72)	<0.001
No (ref)	1 (ref)		1 (ref)	
**Previous history of influenza vaccination**				
Yes	2.05 (1.64–2.55)	<0.001	2.26 (1.74–2.93)	<0.001
No (ref)	1 (ref)		1 (ref)	

**Table 3 vaccines-10-01539-t003:** Associations between chronic conditions and COVID-19 vaccination intention (*n* = 2459).

	Participants Who Are Unwilling to Receive Vaccination (*n* = 384)	Univariable and Multivariable Regression Analysis	*p*
Any physical conditions			
Yes	110 (20.1%)	cOR (95%CI) = 0.66 (0.52–0.85)	0.001
No (ref)	274 (14.3%)	* aOR (95%CI) = 0.68 (0.51–0.90)	0.008
Cardiovascular disease			
Yes	21 (30.4%)	cOR (95%CI) = 0.41 (0.24–0.69)	0.001
No	353 (15.2%)	* aOR (95%CI) = 0.47 (0.26–0.88)	0.018
Hypertension			
Yes	56 (23.6%)	cOR (95%CI) = 0.55 (0.40–0.76)	<0.001
No	315 (14.5%)	* aOR (95%CI) = 0.42 (0.28–0.63)	<0.001
Type 2 diabetes			
Yes	27 (31.8%)	cOR (95%CI) = 0.38 (0.24–0.60)	<0.001
No	346 (14.9%)	* aOR (95%CI) = 0.38 (0.22–0.64)	<0.001
Immunodeficiency			
Yes	41 (39.0%)	cOR (95%CI) = 0.27 (0.18–0.40)	<0.001
No	335 (14.6%)	* aOR (95%CI) = 0.40 (0.24–0.65)	<0.001
Chronic disease of the respiratory system			
Yes	64 (27.2%)	cOR (95%CI) = 0.45 (0.33–0.61)	<0.001
No	311 (14.4%)	* aOR (95%CI) = 0.49 (0.34–0.71)	<0.001
Chronic liver disease			
Yes	33 (55.0%)	cOR (95%CI) = 0.14 (0.08–0.24)	<0.001
No	342 (14.7%)	* aOR (95%CI) = 0.24 (0.13–0.46)	<0.001
Chronic kidney disease			
Yes	33 (57.9%)	cOR (95%CI) = 0.12 (0.07–0.21)	<0.001
No	340 (14.6%)	* aOR (95%CI) = 0.17 (0.09–0.31)	<0.001
Cancer during past 5 years			
Yes	48 (60.8%)	cOR (95%CI) = 0.11 (0.07–0.17)	<0.001
No	327 (14.1%)	* aOR (95%CI) = 0.11 (0.06–0.19)	<0.001
Sickle cell disease			
Yes	33 (67.3%)	cOR (95%CI) = 0.08 (0.05–0.15)	<0.001
No	342 (14.6%)	* aOR (95%CI) = 0.08 (0.04–0.17)	<0.001

* aOR, adjusted for all socio-demographic factors.

**Table 4 vaccines-10-01539-t004:** Associations between mental disorders and COVID-19 vaccination intention (*n* = 2459).

	Participants Who Are Unwilling to Receive Vaccination (*n* = 384)	Univariable and Multivariable Regression Analysis	*p*
Any mental disorders (Any Yes in A35-A47)			
Yes	129 (28.0%)	cOR (95%CI) = 0.38 (0.30–0.48)	<0.001
No (ref)	255 (12.8%)	* aOR (95%CI) = 0.30 (0.22–0.41)	<0.001
Depression			
Yes	86 (28.5%)	cOR (95%CI) = 0.40 (0.31–0.53)	<0.001
No	296 (13.9%)	* aOR (95%CI) = 0.34 (0.24–0.48)	<0.001
Mania/bipolar disorder			
Yes	40 (50.6%)	cOR (95%CI) = 0.16 (0.10–0.25)	<0.001
No	332 (14.2%)	* aOR (95%CI) = 0.16 (0.09–0.27)	<0.001
Psychotic disorders (including schizophrenia)			
Yes	39 (62.9%)	cOR (95%CI) = 0.10 (0.06–0.17)	<0.001
No	339 (14.4%)	* aOR (95%CI) = 0.09 (0.05–0.18)	<0.001
Anxiety disorder			
Yes	53 (23.7%)	cOR (95%CI) = 0.56 (0.40–0.78)	0.001
No	325 (14.8%)	* aOR (95%CI) = 0.50 (0.34–0.74)	<0.001
Posttraumatic stress disorder			
Yes	35 (38.5%)	cOR (95%CI) = 0.28 (0.18–0.43)	<0.001
No	345 (14.8%)	* aOR (95%CI) = 0.31 (0.18–0.51)	<0.001
Eating disorder			
Yes	35 (37.6%)	cOR (95%CI) = 0.29 (0.19–0.44)	<0.001
No	345 (14.8%)	* aOR (95%CI) = 0.31 (0.18–0.52)	<0.001
Compulsive disorders (OCD)			
Yes	55 (50.9%)	cOR (95%CI) = 0.16 (0.11–0.23)	<0.001
No	326 (14.1%)	* aOR (95%CI) = 0.16 (0.09–0.26)	<0.001
Substance abuse or addiction disorder			
Yes	31 (60.8%)	cOR (95%CI) = 0.11 (0.06–0.20)	<0.001
No	349 (14.7%)	* aOR (95%CI) = 0.12 (0.06–0.24)	<0.001
Attention disorder (ADD or ADHD)			
Yes	42 (53.8%)	cOR (95%CI) = 0.14 (0.09–0.23)	<0.001
No	337 (14.3%)	* aOR (95%CI) = 0.15 (0.08–0.26)	<0.001
Somatoform disorder			
Yes	25 (43.1%)	cOR (95%CI) = 0.23 (0.14–0.40)	<0.001
No	356 (15.0%)	* aOR (95%CI) = 0.34 (0.18–0.63)	0.001
Personality disorder			
Yes	42 (66.7%)	cOR (95%CI) = 0.08 (0.05–0.14)	<0.001
No	337 (14.3%)	* aOR (95%CI) = 0.09 (0.04–0.17)	<0.001
Autism Spectrum Disorder			
Yes	33 (62.3%)	cOR (95%CI) = 0.10 (0.06–0.18)	<0.001
No	343 (14.5%)	* aOR (95%CI) = 0.10 (0.05–0.21)	<0.001
Cognitive disorder/dementia			
Yes	32 (74.4%)	cOR (95%CI) = 0.06 (0.03–0.12)	<0.001
No	349 (14.7%)	* aOR (95%CI) = 0.08 (0.04–0.19)	<0.001

* aOR, adjusted for all socio-demographic factors.

**Table 5 vaccines-10-01539-t005:** Associations between perceptions and COVID-19 vaccination intention (*n* = 2459).

	Participants Who Are Unwilling to Receive Vaccination (*n* = 384)	Univariable and Multivariable Regression Analysis	*p*
● Country in which the vaccine is produced.			
Yes	156 (13.9%)	cOR (95% CI) = 1.26 (1.01–1.57)	0.043
No	224 (16.9%)	* aOR (95% CI) = 1.48 (1.15–1.89)	0.002
● Recommendation from my family doctor.			
Yes	126 (8.9%)	cOR (95% CI) = 3.29 (2.61–4.14)	<0.001
No	251 (24.4%)	* aOR (95% CI) = 3.47 (2.67–4.50)	<0.001
● Recommendation of the Ministry of Health.			
Yes	152 (9.5%)	cOR (95% CI) = 3.47 (2.77–4.35)	<0.001
No	229 (26.8%)	* aOR (95% CI) = 3.98 (3.07–5.16)	<0.001
● Whether the vaccine has been in use for 2 years or more.			
Yes	188 (16.7%)	cOR (95% CI) = 0.82 (0.66–1.02)	0.081
No	184 (14.1%)	* aOR (95% CI) = 0.88 (0.69–1.13)	0.308
● Whether the vaccine has no serious side-effects.			
Yes	226 (13.9%)	cOR (95% CI) = 1.45 (1.16–1.81)	0.001
No	155 (18.9%)	* aOR (95% CI) = 1.44 (1.12–1.86)	0.004
● Whether the vaccine is used in other countries.			
Yes	180 (12.1%)	cOR (95% CI) = 1.91 (1.53–2.38)	<0.001
No	199 (20.8%)	* aOR (95% CI) = 1.79 (1.40–2.29)	<0.001
● My risk of getting infected with COVID-19.			
Yes	176 (12.0%)	cOR (95% CI) = 1.93 (1.55–2.41)	<0.001
No	200 (20.8%)	* aOR (95% CI) = 1.95 (1.52–2.51)	<0.001
● How easy it is to get the vaccine (e.g., available out-of-hours or in pharmacies).			
Yes	136 (10.0%)	cOR (95% CI) = 2.65 (2.11–3.33)	<0.001
No	244 (22.7%)	* aOR (95% CI) = 2.78 (2.15–3.59)	<0.001
● Whether the vaccine is free of charge.			
Yes	147 (9.8%)	cOR (95% CI) = 3.04 (2.42–3.81)	<0.001
No	233 (24.8%)	* aOR (95% CI) = 3.04 (2.37–3.91)	<0.001
● Whether restrictions on movement and gathering in groups would be lifted if most people got the vaccine.			
Yes	145 (10.7%)	cOR (95% CI) = 2.32 (1.85–2.90)	<0.001
No	235 (21.7%)	* aOR (95% CI) = 2.32 (1.81–2.98)	<0.001

* aOR, adjusted for all socio-demographic factors.

## Data Availability

The data presented in this study are available on request from the corresponding author. The data are not publicly available due to privacy restrictions. All data that were analysed during this study are included in this published article.

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
