# Peer review of "Factors Associated with Vaccination Intention against the COVID-19 Pandemic: A Global Population-Based Study"

_vaccines, 2022, doi:10.3390/vaccines10091539_

Round 1
Reviewer 1 Report
The Junjie Huang et al paper well describes the factors associated with vaccination intention against the COVID-19 pandemic, considering more than 2.400 individuals across 26 study sites from Middle East, Europe, Americas and Asia-Pacific regions.
Tables are quite explanatory, even if the presented data do not include informations concerning the geographical origin, the style of the local communication and the organization of the national health system of the enrolled participants. It is a basic data: it must be added.
Author Response
The Junjie Huang et al paper well describes the factors associated with vaccination intention against the COVID-19 pandemic, considering more than 2.400 individuals across 26 study sites from Middle East, Europe, Americas and Asia-Pacific regions.
Authors’ response: We are grateful for the positive comment.
Tables are quite explanatory, even if the presented data do not include informations concerning the geographical origin, the style of the local communication and the organization of the national health system of the enrolled participants. It is a basic data: it must be added.
Authors’ response: Thanks for the comment. We have now included a table to list the number of respondents from each country in appendix-supplementary table 1.
The results part has been enhanced, it reads, ‘The numbers of respondents from each country are listed in Appendix-supplementary table 1.’ (Line 157-158)
Reviewer 2 Report
Interesting and well writted paper. The principle of the study is relatively simple, but the large size of the population and the diversity of the patients surveyed make the results relatively robust. I have a few minor comments before consideration for publication:
L65: it would be appropriate to speak of vaccines, in the plural, since several are now on the market.
- the study population appears to be predominantly of Asian origin. It would be interesting to have summary data on the proportion of individuals from the different continents/regions included in the study. Indeed, the reader cannot be misled and must know if the collected data are applicable to the different populations included. Did the authors observe differences in vaccine intention between regions?
- Did the researchers assess whether vaccine intention differed between mRNA and "conventional" vaccines? If they did not, it would be interesting to bring this aspect into the discussion, possibly based on previously published data.
- The researchers should show the questionnaire that was sent out as supplementary material to the article.
Author Response
Interesting and well writted paper. The principle of the study is relatively simple, but the large size of the population and the diversity of the patients surveyed make the results relatively robust.
Authors’ response: We are grateful for the positive comment.
I have a few minor comments before consideration for publication:
L65: it would be appropriate to speak of vaccines, in the plural, since several are now on the market.
Authors’ response: Thanks for the comment. The statement has been revised, ‘The health and pharmaceutical sectors developed effective vaccines to reduce incidence and disease severity, especially in vulnerable populations with chronic conditions [4]. The COVID-19 vaccines haves been found to be effective in minimising the effects of COVID-19 by strengthening the immune system [5]. (Line 65-68)
- the study population appears to be predominantly of Asian origin. It would be interesting to have summary data on the proportion of individuals from the different continents/regions included in the study. Indeed, the reader cannot be misled and must know if the collected data are applicable to the different populations included. Did the authors observe differences in vaccine intention between regions?
Authors’ response: Thanks for the comment. We have now included a table to list the number of respondents from each country in appendix-supplementary table 1.
The results part has been enhanced, it reads, ‘The numbers of respondents from each country are listed in Appendix-supplementary table 1.’ (Line 157-158)
We have conducted an additional analysis to explore the difference in vaccine intention between regions, regression findings showed that there was no significant difference between regions. The result has been included in appendix-supplementary table 2.
The results part has been enhanced, it reads, ‘Additional analysis has been done to explore the difference in vaccine intention between regions, no significant difference has been found (Appendix-supplementary table 2)’ Line 186-188
- Did the researchers assess whether vaccine intention differed between mRNA and "conventional" vaccines? If they did not, it would be interesting to bring this aspect into the discussion, possibly based on previously published data.
Authors’ response: Thanks for the comment. Previous literature has explored the difference in vaccine intention between mRNA and conventional vaccines, it was found that the novelty of mRNA technology would reduce the acceptability of the vaccine. However, such vaccine hesitancy would be reduced with social conformity [24]. (Line 316-319)
Ref: Leong C, Jin L, Kim D, Kim J, Teo YY, Ho T-H. Assessing the impact of novelty and conformity on hesitancy towards COVID-19 vaccines using mRNA technology. Communications Medicine 2022; 2(1): 61.
- The researchers should show the questionnaire that was sent out as supplementary material to the article.
Authors’ response: Thanks for the comment. The questionnaire is now included as a supplementary material in the Appendix.
The following statement has been added to the survey instrument part: The full questionnaire can be found in Appendix-supplementary material 1. (Line 132-133)